

# Initial shifts in nitrogen impact on ecosystem carbon fluxes in an alpine meadow: patterns and causes

Bing Song[1,2], Jian Sun[1], Qingping Zhou[3], Ning Zong[1], Shuli Niu[1,*]

[1]Key Laboratory of Ecosystem Network Observation and Modeling, Institute of Geographic Sciences and Natural Resources Research, Chinese Academy of Sciences, Datun Road, Beijing 100101, China
[2]University of Chinese Academy of Sciences, Yuquan Road, Beijing 100049, China
[3]Institute of Qinghai-Tibetan Plateau, Southwest University for Nationalities, Chengdu, 610041, China
* *Correspondence to*: Shuli Niu (sniu@igsnrr.ac.cn)

## Abstract

The rising nitrogen (N) deposition could increase ecosystem net carbon (C) sequestration by stimulating plant productivity. However, how net ecosystem $CO_2$ exchange (NEE) and its components respond dynamically to rising N deposition is far from clear. Using an N addition gradient experiment (six levels: 0, 2, 4, 8, 16, 32 gN $m^{-2}$ $year^{-1}$) in an alpine meadow on the Tibetan Plateau, we explored the responses of different ecosystem C fluxes to an increasing N loading gradient and revealed mechanisms underlying the dynamic responses. Results showed that NEE, ecosystem respiration (ER), and gross ecosystem production (GEP) all increased linearly with N addition rates in the first year of treatment, but shifted to N saturation responses in the second year with the highest NEE (-7.77 $\pm$ 0.48 $\mu$mol $m^{-2}$ $s^{-1}$) occurring under N addition rate of 8 gN $m^{-2}$ $year^{-1}$. The saturation responses of NEE and GEP were caused by N-induced accumulation of standing litter, which limited light availability for plant growth, under high N addition. The saturation response of ER was mainly due to decreases in aboveground plant respiration and soil microbial respiration under high N addition, while the N-induced reduction in soil pH caused declines in soil microbial respiration. We also found that various components of ER, including aboveground plant respiration, soil respiration, root respiration, and microbial respiration, responded differentially to the N addition gradient. The results reveal temporal dynamics of N impacts and the rapid

shift of ecosystem C cycle from N limitation to N saturation. These findings are helpful for better understanding and model projection of future terrestrial C sequestration under rising N deposition.

## 1 Introduction

Anthropogenic reactive nitrogen (N) inputs to the terrestrial biosphere has increased more than threefold over the past century and is predicted to increase further (Lamarque et al., 2005; Galloway et al., 2008). Because of the strong coupling of ecosystem carbon (C) and N cycles, excess N deposition could have significant impacts on ecosystem C cycle (LeBauer and Treseder, 2008; Liu and Greaver, 2010; Lu et al., 2011). Ecosystem net C sequestration is usually predicted to increase under rising N deposition (Magnani

et al., 2007; Reay et al., 2008; Fernandez-Martinez et al., 2014). However, we have limited understanding on the dynamics of N regulations of C sequestration in terrestrial ecosystems, which is crucial for model projection of future terrestrial C cycle under rising N deposition (Reay et al., 2008).

Although N addition generally enhances plant growth, ecosystem net primary productivity (NPP), and plant C pool size according to global syntheses of N addition experiments (LeBauer and Treseder,

2008; Xia and Wan, 2008; Lu et al., 2011), N responses of ecosystem C fluxes vary with N loading rates (Liu and Greaver, 2010; Lu et al., 2011). Low rates of N addition could stimulate ecosystem productivity (Aber et al., 1989) and soil respiration (Hasselquist et al., 2012; Zhu et al., 2016), but high doses of N addition could have negative effects on soil respiration and microbial biomass (Treseder, 2008; Janssens et al., 2010; Maaroufi et al., 2015). Therefore, we need multi-levels of N loading to explicitly quantify

the dynamic responses of ecosystem C cycle to increasing N loading. According to N saturation theory, NPP was assumed to slowly increase with N addition rates first, then get at the maximum value at N

saturation point and finally decline with further increase of N input (Aber et al., 1989; Lovett and Goodale, 2011). However, other C cycle processes have not yet been well examined against N saturation concept. In addition, the N response of ecosystem C fluxes may shift with time because of changes in plant community structure and other limiting factors (Niu et al., 2010). We don't know the time when ecosystem

C cycle gets N saturated yet. The mechanisms underlying why C fluxes get saturated with N inputs are also even far from clear, which hinders us from accurately predicting the C cycle in response to rising N deposition.

Large uncertainty also remains for the direction, magnitude, and underlying mechanisms of different C flux components in response to increasing N loading. Net ecosystem $CO_2$ exchange (NEE) is the

balance between gross ecosystem production (GEP) and ecosystem respiration (ER) (Chapin III et al., 2011). Limited number of previous studies demonstrated that NEE had a positive (Niu et al., 2010; Huff et al., 2015) or no significant response (Bubier et al., 2007; Harpole et al., 2007) to N addition. The insignificant changes in NEE may be due to the incongruent responses of GEP and ER to N addition. GEP is determined by plant photosynthesis, while ER can be divided into aboveground plant respiration,

belowground plant respiration (root respiration), and soil microbial respiration. These components of ER could be affected by plant aboveground biomass, root biomass, soil organic matter and microbial biomass C, respectively, and may respond variously to N addition (Phillips and Fahey, 2007; Hasselquist et al., 2012). For example, root respiration would be enhanced or not significantly changed under N addition, while soil microbial respiration may be suppressed by N addition (Zhou et al., 2014). The different

responses of various components of ecosystem C cycle to N addition will consequently change NEE. Nevertheless, there is still limited knowledge on how various components of NEE respond differentially

to N addition gradient.

In this study, we explored the responses of various ecosystem C cycle processes to an N addition gradient in an alpine meadow on the Qinghai-Tibetan Plateau. The Qinghai-Tibetan Plateau has an area of 2.5 million km$^2$ with alpine meadow covering 35 % of this area, and it is sensitive to environmental

change and human activities (Chen et al., 2013). The objectives of this study were to explore how different components of ecosystem C fluxes respond to increasing N loading gradient. Specifically, we addressed the following questions: (i) how do NEE and its components respond to N addition gradient in the alpine meadow? (ii) whether various C cycle processes can get N saturated? If so, at which N addition level they are saturated and how do the responses shift with time? and (iii) what are the mechanisms underlying N

saturation responses of different C cycle processes?

## 2 Materials and methods

### 2.1 Study site

The study site is located in an alpine meadow in Hongyuan County, Sichuan Province, China, which is

on the eastern Qinghai-Tibetan Plateau (32 °48′ N, 102 °33′ E). The altitude is ~3500 m. Long-term (1961-2013) mean annual precipitation is 747 mm with approximately 80 % occurring in May to September. Long-term mean annual temperature is 1.5 ℃ with monthly mean temperature ranging from -9.7 ℃ in January to 11.1 ℃ in July. The dominant species in this alpine meadow are *Deschampsia caespitosa* (Linn.) Beauv., *Kobresia setchwanensis* Hand. -Mazz., *Carex schneideri* Nelmes, and *Anemone rivularis*

Buch.-Ham.. The vegetation cover of this grassland is over 90 %. The soil in the study site is classified as Mat Cry-gelic Cambisol according to the Chinese classification, with surface soil bulk density being

0.89 g cm$^{-3}$. The soil organic C content and total N content are 37 gC kg$^{-1}$ and 3.5 gN kg$^{-1}$, respectively.

The background N deposition is ranging from 0.87 to 1.38 gN m$^{-2}$ year$^{-1}$ on the eastern Qinghai-Tibetan

Plateau (Lü and Tian, 2007).

## 2.2 Experimental design

We conducted an N addition experiment with six levels of N addition rate (0, 2, 4, 8, 16, 32 gN m$^{-2}$ year$^{-1}$) in early 2014. The six N treatments were represented by N0 (control), N2, N4, N8, N16, and N32, respectively. The treatments were randomly assigned with five replications, so there were totally 30 plots. Each plot was 8×8 m, and the distance between any two adjacent plots was 3 m. The N addition treatments started from May, 2014. In 2014 and 2015, N was applied by hand as $NH_4NO_3$ (>99 %) every month from May to September (i.e. during the growing season) before rainfall. In order to distribute dry $NH_4NO_3$ evenly in the plots, we mixed dry $NH_4NO_3$ with enough amounts of soil to apply.

## 2.3 Ecosystem C cycle properties and soil pH measurement

Ecosystem C fluxes were measured using a transparent static chamber (0.5×0.5×0.5 m) attached to an infrared gas analyzer (LI-6400XT; LI-COR Environmental, Lincoln, Nebraska, USA) in the field. During each measurement, the chamber was positioned over a square steel frame, which was permanently inserted into soil and offered a flat base for the chamber. Inside the chamber, two electric fans were mounted in order to mix the chamber atmosphere. The measurements were conducted twice per month on clear, sunny days from May to September in 2014 and 2015. Nine consecutive recordings of $CO_2$ concentration were taken on each base at 10-second intervals. $CO_2$ flux rates were determined from the

time-courses of the concentrations to calculate net ecosystem $CO_2$ exchange (NEE). After the measurement of NEE, the chamber was covered by an opaque cloth and the $CO_2$ measurement was repeated. As the second measurement eliminated light, the $CO_2$ flux value obtained represented ecosystem respiration (ER). Gross ecosystem production (GEP) was calculated as the difference between NEE and

ER. Negative or positive NEE and GEP values represent net C uptake or release, respectively. The detailed methods have also been described in Niu et al. (2008) and Niu et al. (2013).

Soil respiration (SR) was assessed following the measurement of NEE and ER. It was also measured with LI-6400XT attaching a soil $CO_2$ flux chamber (991 cm$^3$ in total volume; LI-6400-09; LI-COR Environmental, Lincoln, Nebraska, USA). A PVC collar (10.5 cm in diameter and 5 cm in height) was

permanently installed 2-3 cm into the soil. The soil respiration chamber attached to LI-6400XT was placed on each PVC collar for 1-2 minutes to measure SR. Living plants inside the collars were removed regularly by hand to eliminate aboveground plant respiration. Soil heterotrophic respiration (i.e. soil microbial respiration, $R_{mic}$) was measured using the same method as soil respiration. Differently, the PVC collar was 40 cm in height and installed 36-38 cm into the soil. As >90 % of plant roots were distributing

in the topsoil (0-20 cm), 40-cm-long PVC collars could cut off old plant roots and prevented new roots from growing inside the collars. Plants in the collars were completely removed by hand to exclude C supply. The experiment was conducted in early 2014 and the measurements of $CO_2$ fluxes above these 40-cm-long PVC collars began in late July in 2014, leaving enough time for the remaining plant roots inside the collars to die. Thus $CO_2$ fluxes in those deep collars represented $R_{mic}$. The method was same to

Wan et al. (2005) and Zhou et al. (2007). Root respiration ($R_{root}$) was calculated by value of SR minus $R_{mic}$. Aboveground plant respiration ($R_{above}$) was calculated by ER minus SR, and ecosystem plant

respiration ($R_{plant}$) was calculated as the difference between ER and $R_{mic}$. All the measurements of ecosystem C fluxes were contemporaneous.

Soil samples were collected from the topsoil (0-10 cm) of the 30 plots on August 15, 2014 and August 14, 2015. Two soil cores (8 cm in diameter and 10 cm in depth) were taken at least 1 m from the edge in each plot, and then completely mixed to get a composite sample. The soil samples were sieved by a 2 mm mesh and then were air-dried for chemical analysis. Soil pH was determined with a glass electrode in a 1:2.5 soil:water solution (w/v).

## 2.4 Statistical analysis

Repeated-measures ANOVA (RMANOVA) was used to examine N addition effects on each ecosystem C flux over the growing season in 2014 and 2015. When we evaluate N addition effects on the different components of ER and their proportions, we averaged their values across the year and then used one-way ANOVA to test the differences among treatments. To test the response pattern of ecosystem C cycle properties to the N addition gradient, we fitted the response parameter to linear or quadratic functions which had the highest $R^2$. Simple linear regression analyses were used to evaluate relationships of ER with its components and NEE across the two years. $\Delta R_{mic}$ and $\Delta pH$ were calculated by data in different N addition treatments minus data in the control treatment. All data were tested for normal distribution before statistical analysis. The a posteriori comparisons were performed by DUNCAN test, and the effects were considered to be significantly different if $P<0.05$. All statistical analyses were conducted with SAS V.8.1 software (SAS Institute Inc., Cary, North Carolina, USA).

**BioGeosciences**
Discussions

# 3 Results

## 3.1 NEE and its components in response to N addition gradient

Net ecosystem $CO_2$ exchange (NEE) varied throughout the growing seasons in both 2014 and 2015. The maximum rates of net $CO_2$ uptake (indicated by large negative values of NEE) occurred in July in both years (Fig. 1a,d). N addition had a significant impact on NEE in 2014 ($P$=0.020) and a marginally significant effect in 2015 ($P$=0.059) (Table 1). Annual mean NEE had different responses to the N addition gradient between the two years (Fig. 1a,d). It increased linearly with N addition rates in 2014 (Fig. 2a), but shifted to a saturating response with N addition rates in 2015 (Fig. 2d). The largest NEE was -7.77 ± 0.48 µmol m$^{-2}$ s$^{-1}$ under 8 gN m$^{-2}$ year$^{-1}$ addition rate (N8) in 2015.

The N addition gradient had significant effects on ER ($P$=0.033 and 0.006, respectively) and GEP ($P$=0.002 and 0.038, respectively) in both 2014 and 2015 (Table 1). Similar to NEE, both ER and GEP showed linear responses to N addition rates in 2014 but shifted to saturation responses in 2015 (Fig. 2). On average, ER was enhanced by 0.9-16.1 % in 2014 and 7.9-23.7 % in 2015 under different N addition treatments. GEP was increased by 2.4-19.2 % in 2014 and 6.7-20.5 % in 2015 under different N addition levels, with maximal values being -24.40 ±0.48 µmol m$^{-2}$ s$^{-1}$ under 32 gN m$^{-2}$ year$^{-1}$ in 2014 and -15.38 ±0.72 µmol m$^{-2}$ s$^{-1}$ under 16 gN m$^{-2}$ year$^{-1}$ in 2015 (Fig. 2).

## 3.2 Components of ecosystem respiration in response to N addition gradient

We divided ER into aboveground plant respiration ($R_{above}$), soil respiration (SR), root respiration ($R_{root}$), and microbial respiration ($R_{mic}$), and found that different ER components showed diverse responses to N addition gradient. Annual mean SR was not significantly changed by N addition gradient in 2014 (Table

1; Fig. 3). However, in 2015, it ranged from 4.98 $\pm$ 0.33 $\mu$mol m$^{-2}$ s$^{-1}$ to 6.23 $\pm$ 0.23 $\mu$mol m$^{-2}$ s$^{-1}$ under different N addition levels, with significant reduction under high N addition levels of 16 and 32 gN m$^{-2}$ year$^{-1}$ ($P$=0.010; Fig. 3). Additionally, the relationship between SR and N addition rates was not significant in 2014 (Fig. 3a), while SR leveled off under high N addition rates in 2015 (Fig. 3c).

Interestingly, $R_{mic}$ increased linearly with N addition rates in 2014 (Fig. 3b), while it decreased with N addition rates in 2015 (Fig. 3d).

      $R_{above}$ increased with increasing N addition rates in 2014 (Fig. 4b) but got the maximum value at N16 in 2015 (Fig. 4e). By contrast, $R_{root}$ decreased with increasing N addition rates in 2014 (Fig. 4c), while it had no statistically significant response to N addition gradient in 2015 (Fig. 4f). Comparing

among various components of ER, only $R_{mic}$ showed distinctively inverse responses to N addition rates between years, which kept increasing in 2014 but decreasing in 2015 along the N addition gradient (Fig. 3). All other components of ER generally showed similar response tendency between two years (Fig. 3a,3c,4).

      In addition, the proportions of different efflux components to ER differed in response to N addition

gradient between years (Fig. 5). The proportions of $R_{above}$ to ER kept increasing with N addition rates in 2014 but got saturated at N16 in 2015 (Fig. 5a,d). The proportions of $R_{root}$ to ER ranged from 31.90 $\pm$ 6.69 % in N0 plots to 11.18 $\pm$ 1.28 % in N32 plots in 2014 (Fig. 5b), but was not significantly different among N addition levels in 2015 (Table 1; Fig. 5e). In 2014, the contributions of $R_{mic}$ to ER did not significantly change under N addition treatments (Table 1; Fig. 5c), whereas they declined along the N

addition gradient in 2015 (Fig. 5f).

### 3.3 Causes for the N saturation responses of ecosystem C fluxes

In order to examine the causes for the N saturation responses of ER in 2015, we examined the relationship between ER and its various components. The results showed that ER had significantly positive correlation with $R_{above}$ and $R_{mic}$ (Fig. 6a,c) but not with $R_{root}$ (Fig. 6b). Moreover, NEE closely correlated with ER (Fig. 6d). The findings indicated that the saturation response of $R_{above}$ and the declined response of $R_{mic}$ in combination contributed to N saturation response of ER and the consequent saturation response of NEE in 2015. We further explored the causes for decreasing $R_{mic}$ with N addition in 2015 and found that N addition significantly reduced soil pH in 2015 (Fig. 7a). N-induced reduction in soil microbial respiration ($\triangle R_{mic}$) was positively dependent on N-induced reduction in soil pH ($\triangle pH$) in 2015 (Fig. 7b). The decreased $R_{above}$ at high N addition rates was attributed to the accumulated standing litter mass and thus less light condition under high N addition treatments (Fig. S1).

### 4 Discussion

### 4.1 Nitrogen saturation responses of ecosystem C fluxes and the causes

Our findings showed that ecosystem C fluxes (NEE, ER, and GEP) had linear responses in the first year but shifted to saturation responses in the second year (Fig. 2,3). The linear responses in the first year suggest N limitation for ecosystem C cycle in this alpine meadow. The saturation responses in the second year indicate N demands for ecosystem C fluxes may get saturated under high N addition rates and will decrease further with more N addition. Being consistent with the N saturation hypothesis proposed for the response of NPP to N addition (Aber et al., 1998; Aber et al., 1989; Lovett and Goodale, 2011), NEE also showed three stages of response to N gradient in the second year, in which net C sequestration

increased first with N addition levels, then saturated under N addition rate of approximately 8 gN m$^{-2}$ year$^{-1}$, and eventually decreased with any higher N addition rates. Most previous N addition studies used only one level of N addition and found that NEE showed a positive (Niu et al., 2010; Huff et al., 2015) or no significant response (Harpole et al., 2007; Bubier et al., 2007) to N addition. One level of N addition could not give solid assessment and quantification of ecosystem responses to N addition. By using an N addition gradient experiment, this study comprehensively showed the dynamic responses of NEE and its components to different N loading rates.

The N saturation responses of ER and thus NEE are mainly caused by the decrease of aboveground plant respiration and soil microbial respiration under high N addition treatments in 2015. The decrease of aboveground plant respiration under N32 treatment is primarily due to that N addition stimulated plant growth and thus standing litter accumulation after plant senescence (Fig. S1). The greater standing litter mass could reduce light availability for plant growth under N32 treatment in the second year. Therefore, NEE decreased at the highest N addition rate, leading to N saturation response. The N-induced light limitation for plant growth was also observed in other ecosystems, like temperate grassland (Niu et al., 2010; Kim and Henry, 2013). Moreover, our results showed that most components of ER had similar response patterns between the two years except soil microbial respiration that increased in 2014 but decreased in 2015 along with N addition rates. The relationships between ER and soil microbial respiration (Fig. 6c) indicate that the decrease of microbial respiration contributes to the reduction of ER under high N addition rates in 2015. The decline of microbial respiration under high N addition conditions was primarily due to the N-induced reduction in soil pH (Fig. 7). Previous study suggested soil pH was the most important driver for responses of microbes under high N addition rates (Liu et al., 2014; Song

et al., 2014). N addition can lead to soil acidification and bring negative impacts on soil microbial growth

and activities (Liu et al., 2014; Tian et al., 2016). In this study, the decreased soil pH may cause toxicity

effects on microbial activity (Treseder, 2008; Zhou et al., 2012) and thus reduces microbial respiration

after two years of N addition.

## 4.2 The time and N threshold for the saturation responses

Our findings demonstrate that N responses of ecosystem C fluxes shifted from linear response to

saturation response over the two years of treatments (Fig. 2). A recent study revealed that ecosystem C

fluxes exhibited saturating responses to N addition during two consecutive measurement years in a

temperate grassland (Tian et al., 2016). However, their measurement was conducted after ten years of N

addition treatments (similar N addition rates with our study), so it did not capture the early response

signals of ecosystem C exchange. Results of another N addition gradient experiment carried out in three

marsh ecosystems showed that aboveground plant biomass increased linearly with N addition rates after

seven months of treatment, but showed saturating responses after 14 months of N addition (Vivanco et

al., 2015). Taken together with our results, it suggests that N saturation of ecosystem C fluxes may happen

very quickly.

The N saturation threshold for ecosystem C fluxes of this alpine meadow is approximately 8 gN m$^{-2}$ year$^{-1}$. This level is much higher than that in an alpine steppe on the Qinghai-Tibetan Plateau (Liu et al.,

2013). In Liu et al.'s study, biomass N concentration, soil N$_2$O flux, N-uptake efficiency and N-use

efficiency showed saturating responses at N addition rate of 4 gN m$^{-2}$ year$^{-1}$. The discrepancy is probably

caused by different precipitation at the two sites. The precipitation is 747 mm in our study site and is 415

mm in their study site. The lower precipitation may constrain ecosystem's response to N addition. Likewise, the N saturation load for our alpine meadow is higher than that for an alpine dry meadow in Colorado (Bowman et al., 2006) and is comparable with a temperate steppe of Eurasian grasslands with a saturable N addition rate of approximately 10.5 gN m$^{-2}$ year$^{-1}$ (Bai et al., 2010). The higher saturation

levels indicate that this alpine meadow is more limited by N comparing with other resources. Furthermore, the N critical load for causing changes in ecosystem C cycle processes is around 2 gN m$^{-2}$ year$^{-1}$ in this alpine meadow. In the first year, ecosystem C exchanges were not significantly different between N0 and N2 treatments, but C fluxes were greater in N2 plots than that in N0 plots in the second year (Fig. 1). This threshold for triggering ecosystem C fluxes changes is comparable to that in another alpine meadow on

the mid-south of the Tibetan Plateau (Zong et al., 2016). Observed atmospheric wet N deposition is ranging from 0.87 to 1.38 gN m$^{-2}$ year$^{-1}$ on the eastern Qinghai-Tibetan Plateau (Lü and Tian, 2007). Our estimate on N critical load suggests that ecosystem C cycle would be largely affected under future N deposition scenarios and ecosystem may sequester more C from the atmosphere in the alpine meadow of Qinghai-Tibetan Plateau.

### 4.3 Diverse responses of C flux components to N addition gradient

The components of ER showed diverse responses to the N addition gradient (Fig. 4,5). For example, in 2014, aboveground plant respiration and its proportion to ER increased, but belowground plant respiration and its proportion to ER decreased with N addition amounts. Microbial respiration decreased and its

proportion to ER did not change with the N addition gradient. To our knowledge, there was no previous study examining different components of ER in response to N addition gradient. Previous studies

conducted in alpine grassland demonstrated that N addition had no significant effects on ER (Jiang et al., 2013; Gong et al., 2014), since aboveground biomass did not respond to N addition in their studies. In this study, greater plant growth and aboveground biomass under N addition enhanced aboveground plant respiration and then stimulated ER. The lack of N effect on soil respiration (SR) in 2014 may be attributed

to the counteractive responses of soil microbial respiration and root respiration to N addition. In the first year, N addition ameliorated the nutrient limitation for microbes, thus soil microbial activity and biomass increased in short term (Treseder, 2008) and subsequently stimulated microbial respiration (Peng et al., 2011). On the other hand, N addition could reduce belowground biomass allocation (Haynes and Gower, 1995), leading to decrease in root respiration. The increase of soil microbial respiration partly offsets the

decrease of root respiration. As a result, SR had no significant difference among N treatments in the first year. However, in the second year, soil microbial respiration declined under high N addition levels, in combination with the low root respiration, resulting in decreases of SR under N16 and N32 treatments. This decrease in SR was also observed in other ecosystems under long-term or high levels of N addition (Yan et al., 2010; Zhou and Zhang, 2014; Maaroufi et al., 2015). In summary, these results indicate that

ER and its components could respond to N addition gradient in different ways.

## 5 Conclusions

Based on a field N addition gradient experiment, this study tested N saturation theory against multiple C cycle processes and found that ecosystem C fluxes of NEE, GEP, and ER shifted from linear

responses to saturation responses over two years of N addition. The saturation responses of NEE and ER were mainly caused by N-induced decreases in aboveground plant respiration and soil microbial

respiration under high N addition rates. Furthermore, N-induced reduction in soil pH was the main mechanism underlying declines in microbial respiration under high N addition. The N critical load for causing ecosystem C fluxes changes and the N saturation threshold in this alpine meadow were 2 and 8 gN m$^{-2}$ year$^{-1}$, respectively. We also revealed that various components of ER, including aboveground plant respiration, soil respiration, root respiration, and microbial respiration, responded differentially to N addition gradient. The findings suggest that C cycle processes have differential responses to N addition between aboveground and belowground plant parts, and between plants and microbes. Our findings provide experimental evidences for N responses of ecosystem C cycle, which is helpful for parameterizing biogeochemical models and guiding ecosystem management in light of future increasing N deposition.

## Acknowledgements

The authors thank Xiaojing Qin, Yanfang Li, Fangyue Zhang, QuanQuan, Zheng Fu, Qingxiao Yang and Xiaoqiong Huang for their help in field measurement. We thank the staff of Institute of Qinghai-Tibetan Plateau in Southwest University for Nationalities. This study was financially supported by National Natural Science Foundation of China (31625006, 31470528), the Ministry of Science and Technology of China (2016YFC0501803, 2013CB956300), the "Thousand Youth Talents Plan", and West Light Foundation of the Chinese Academy of Sciences.

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





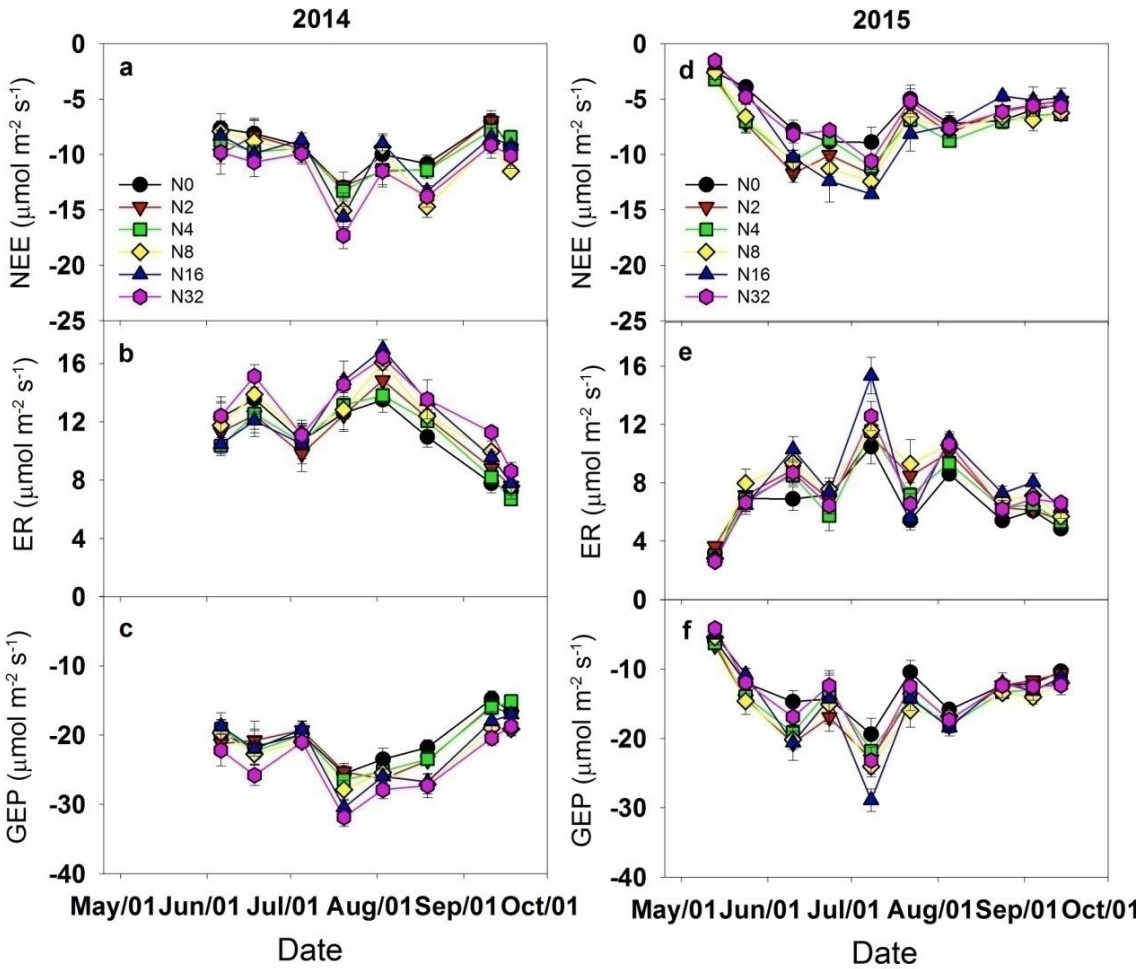

**Figure 1. Seasonal dynamics of net ecosystem CO₂ exchange (NEE) (a, d), ecosystem respiration (ER) (b, e), and gross ecosystem production (GEP) (c, f) in 2014 and 2015. N0, N2, N4, N8, N16, N32 represent N addition rate of 0, 2, 4, 8, 16, 32 gN m⁻² year⁻¹, respectively.**

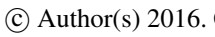


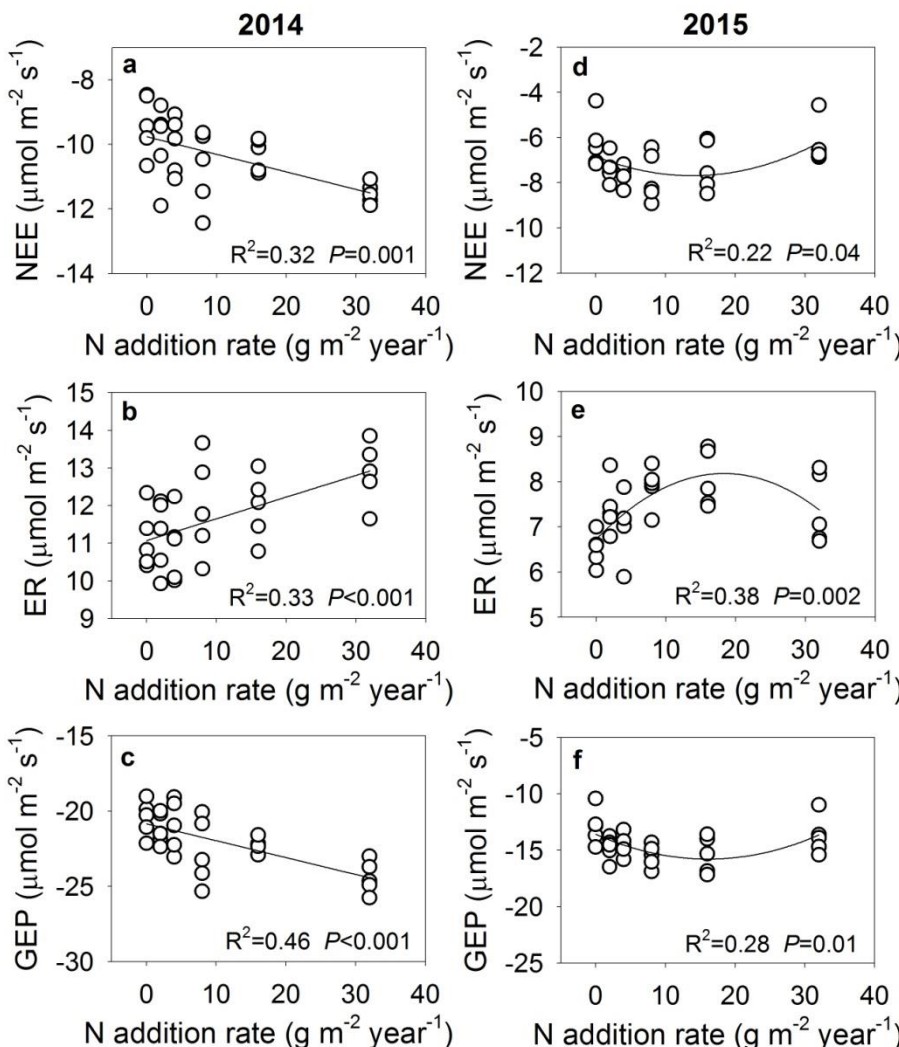

**Figure 2. Relationships between N addition rate and net ecosystem CO₂ exchange (NEE) (a, d), ecosystem respiration (ER) (b, e), and gross ecosystem production (GEP) (c, f) in 2014 and 2015.**



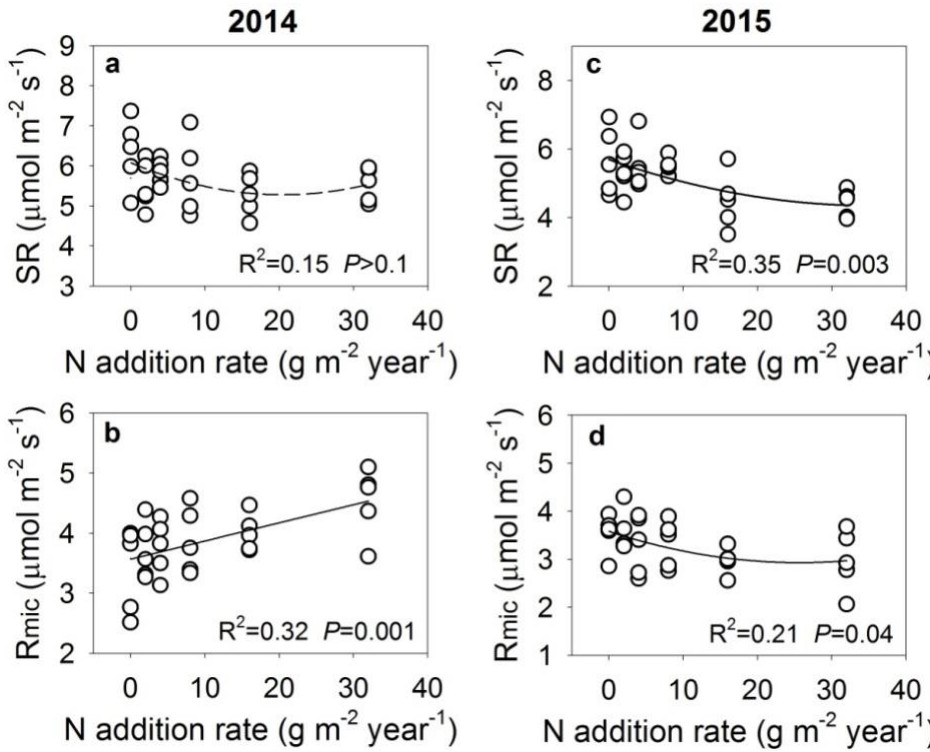

**Figure 3. Relationships between N addition rate and soil respiration (SR) (a, c), and soil microbial respiration ($R_{mic}$) (b, d) in 2014 and 2015.**




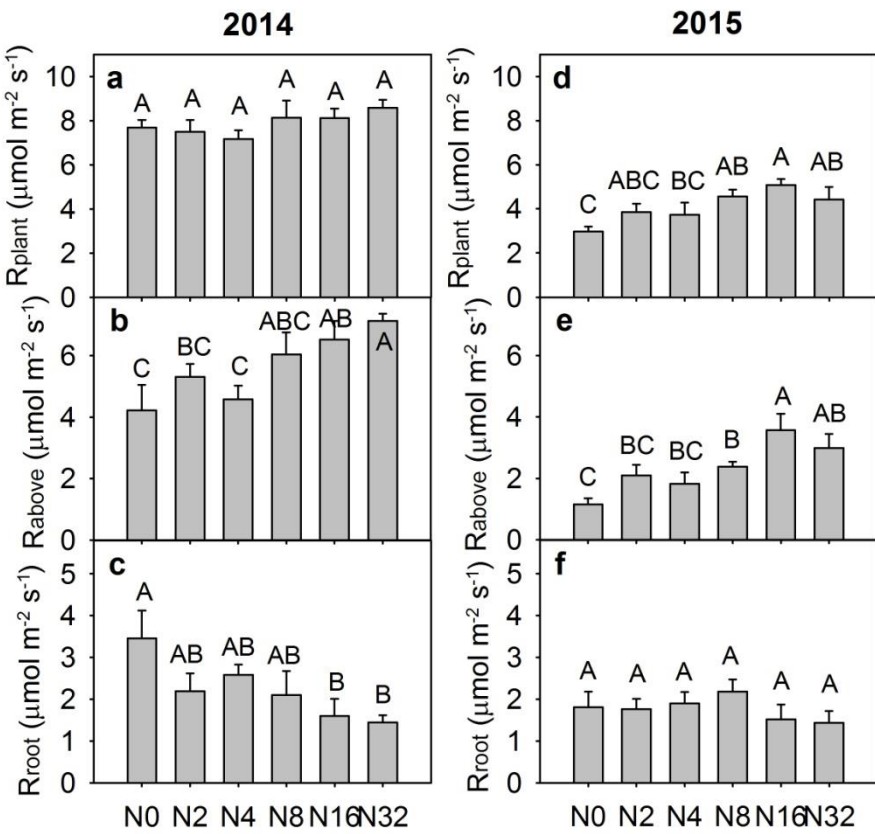

**Figure 4. Plant respiration and its components in response to the N addition gradient in 2014 and 2015 (mean ±SE, n = 5). $R_{plant}$: plant respiration (a, d), $R_{above}$: aboveground plant respiration (b, e), $R_{root}$: plant root respiration (c, f). N0, N2, N4, N8, N16, N32 represent N addition rate of 0, 2, 4, 8, 16, 32 gN m$^{-2}$ year$^{-1}$, respectively.**





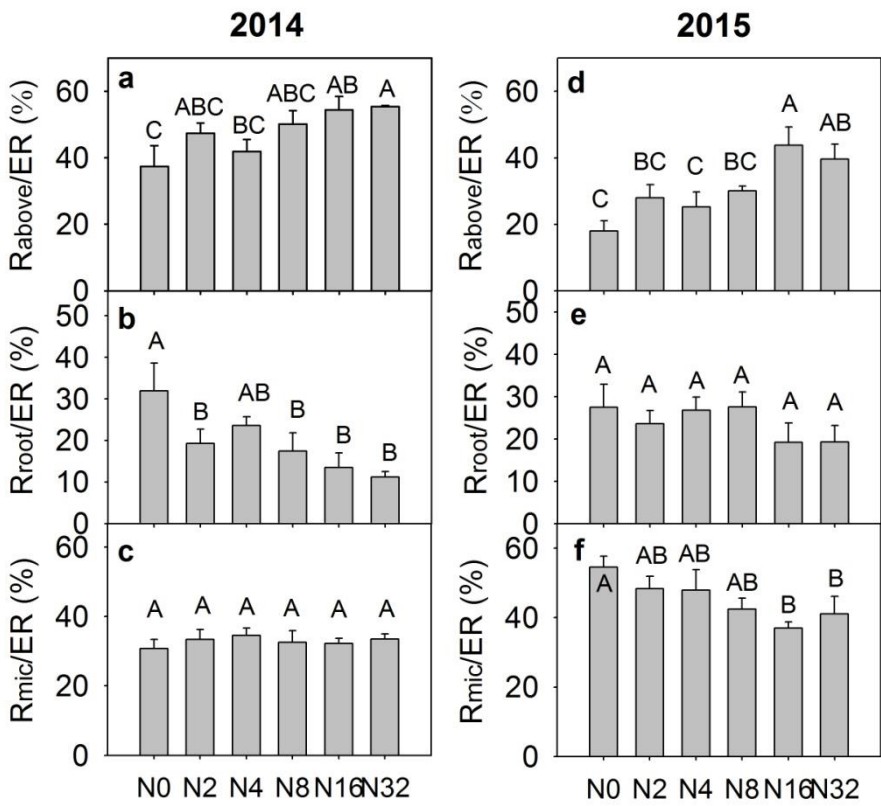

**Figure 5. The contributions of different source components to ecosystem respiration (ER) in response to the N addition gradient in 2014 and 2015 (mean ±SE, n = 5). $R_{above}$: aboveground plant respiration, $R_{root}$: plant root respiration, $R_{mic}$: soil microbial respiration. N0, N2, N4, N8, N16, N32 represent N addition rate of 0, 2, 4, 8, 16, 32 gN m$^{-2}$ year$^{-1}$, respectively.**



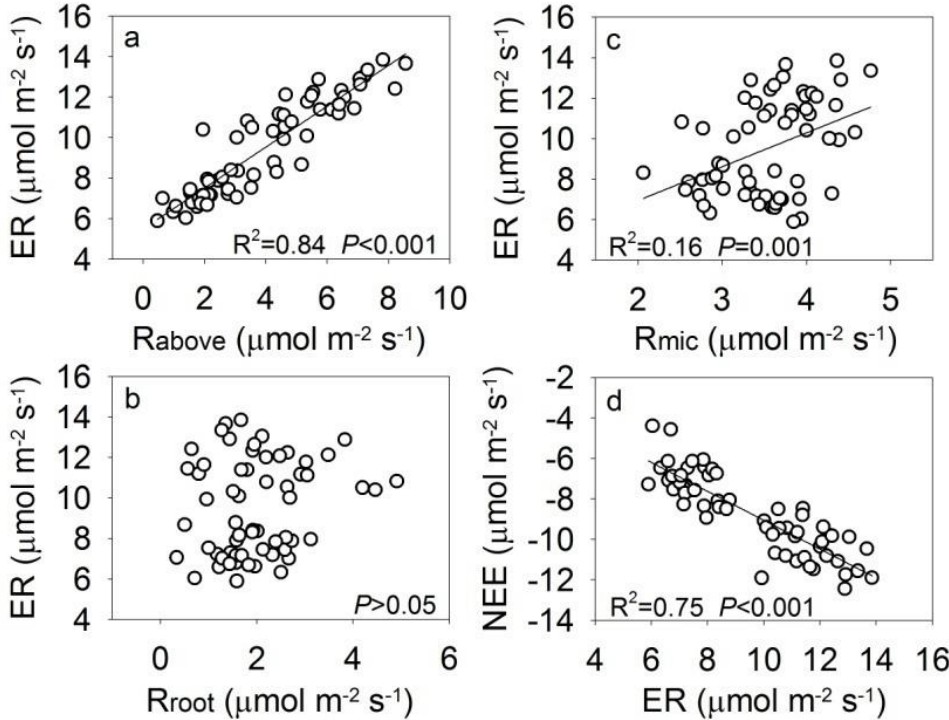

**Figure 6. Relationships between aboveground plant respiration ($R_{above}$), root respiration ($R_{root}$), soil microbial respiration ($R_{mic}$) and ecosystem respiration (ER) (a,b,c), ER and net ecosystem CO$_2$ exchange(NEE) (d) across all plots in 2014 and 2015.**




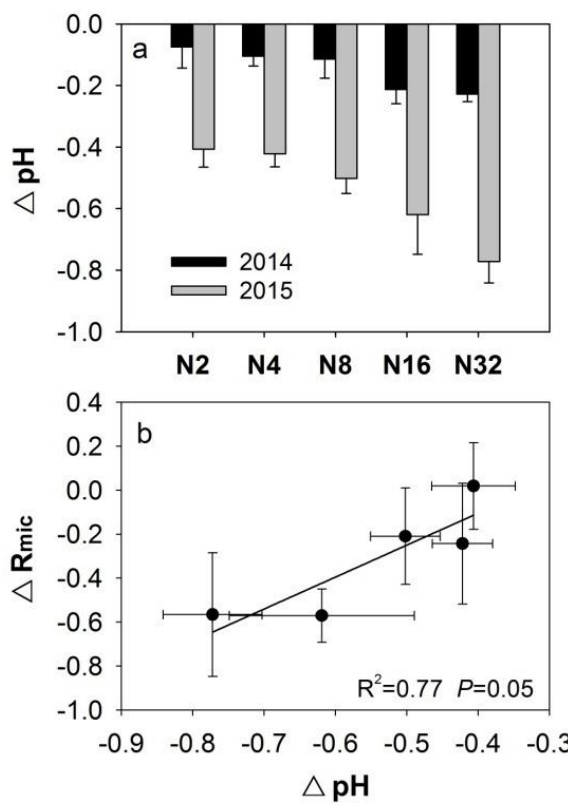

**Figure 7. N-induced changes in soil pH ($\triangle$pH) (a) (mean ± SE, n = 5) and the dependence of N-induced changes in soil microbial respiration ($\triangle R_{mic}$) on N-induced changes in soil pH ($\triangle$pH) in 2015 (b). N2, N4, N8, N16, N32 represent N addition rate of 2, 4, 8, 16, 32 gN m$^{-2}$ year$^{-1}$, respectively.**





**Table 1. Results (F and _P_ values) of one-way ANOVA on the effects of nitrogen addition on ecosystem C fluxes in 2014 and 2015. NEE: net ecosystem $CO_2$ exchange, ER: ecosystem respiration, GEP: gross ecosystem production, SR: soil respiration, $R_{mic}$: soil microbial respiration, $R_{plant}$: plant respiration, $R_{above}$: aboveground plant respiration, $R_{root}$: plant root respiration.**

| | df | NEE | | ER | | GEP | | SR | | $R_{mic}$ | |
|---|---|---|---|---|---|---|---|---|---|---|---|
| | | F | _P_ | F | _P_ | F | _P_ | F | _P_ | F | _P_ |
| 2014 | 5 | 3.35 | 0.020 | 2.95 | 0.033 | 5.37 | 0.002 | 1.56 | 0.209 | 1.49 | 0.246 |
| 2015 | 5 | 2.50 | 0.059 | 4.35 | 0.006 | 2.83 | 0.038 | 3.94 | 0.010 | 1.40 | 0.259 |

| | df | $R_{plant}$ | | $R_{above}$ | | $R_{root}$ | | $R_{above}$/ER | | $R_{root}$/ER | | $R_{mic}$/ER | |
|---|---|---|---|---|---|---|---|---|---|---|---|---|---|
| | | F | _P_ | F | _P_ | F | _P_ | F | F | F | _P_ | F | _P_ |
| 2014 | 5 | 1.06 | 0.409 | 3.84 | 0.011 | 2.64 | 0.049 | 3.08 | 0.027 | 3.56 | 0.015 | 0.28 | 0.919 |
| 2015 | 5 | 3.25 | 0.022 | 5.38 | 0.002 | 0.78 | 0.573 | 5.54 | 0.002 | 0.97 | 0.456 | 2.46 | 0.062 |