# Peer review of "Initial shifts in nitrogen impact on ecosystem carbon fluxes in an alpine meadow: patterns and causes"

_Biogeosciences, 2016_

## Referee Comment (RC1) · Anonymous Referee #1 · 30 Nov 2016

This study addresses potential responses of different ecosystem C fluxes to gradual increases in N fertilization. The main findings of this study provide evidence that N saturation of ecosystem C fluxes can occur in a short period of time (just over 2 years since the start of the N fertilization experiment). Key findings are shown in Fig 2 where differences in NEE and ER are clear between years and along the N fertilization treatment.

Despite the results indicate that N saturation may occur at increasing N fertilization levels, the underlying mechanisms explaining why C fluxes might get saturated with N inputs are not clear. The authors suggest that decreases in NEE and ER under greater N fertilization are due to decreases in plant aboveground respiration and soil microbial respiration. Looking at Figs 3 and 4, this interpretation is not really supported by results whereby plant aboveground respiration (in 2015; Fig 4a) seems to increase rather than

decrease at N8,16,32 treatments compared to N0,2,4. Similarly soil microbial respiration does not seem to decrease much under N8,16,32 treatments (Fig 3d) and actually might increase under N32 compared to N16. My point here is that although NEE and ER trends are relatively clear, the mechanisms invoked here to explain these changes are not really supported by the results. There is a problem with results interpretation here that the authors need to deal with (see my comments below).

I think the authors should either better demonstrate that soil microbial respiration might play a role in mediating the N saturation effect or that other mechanisms are at play. It looks like that soil respiration in general decreases more convincingly under higher N treatments than soil microbial respiration. Also the explanation that greater standing litter might reduce plant aboveground respiration through reduced light availability makes sense but is not really supported by the results in Fig 4e for example.

Also in relation to results interpretation, the authors need to acknowledge that variability in their findings could be related to their very short-term study, which may not capture key changes in NEE and ER and the underlying mechanisms involved. I would expect that it will take 3-5 years of N fertilization to better clarify these.

Overall, the manuscript needs a thorough editing in relation to sentence structure and language especially abstract and introduction but in general all throughout the manuscript.

Discussion

I am not sure whether the explanation that: "The N saturation responses of ER and thus NEE are mainly caused by the decrease of aboveground plant respiration and soil microbial respiration under high N addition treatments in 2015" (page 11, lines 8-9), is well supported by the results. If I look at Fig. 4e I see an increase in aboveground plant respiration (i.e. Rabove) in 2015 under the N16 treatment and a slight decrease under the N32 treatment, which is however still higher than the N8 treatment. What I can see is an overall decrease of Rabove across all treatments in 2015 when compared to

2014. Even the 'assumed' decreases in soil microbial respiration are not clear in Fig. 3d, actually it looks like that Rmic almost increases between N16 and N32.

Page 11, lines 9-11. I might agree with the statement that: "The decrease of above-ground plant respiration under N32 treatment is primarily due to that N addition stimulated plant growth and thus standing litter accumulation after plant senescence (Fig. S1)", but again this is not clear from the results shown. Fig. S1 might provide evidence of litter accumulation but is this the only treatment (N32), which was associated with an increase of plant litter? What about N16?

Again on pag. 11, lines 17-19, the authors suggest that: "The relationships between ER and soil microbial respiration (Fig. 6c) indicate that the decrease of microbial respiration contributes to the reduction of ER under high N addition rates in 2015", which is not really what is shown in Fig. 6c. This figure shows an overall positive relationship between Rmic and ER but this has not to do with increases in N addition rates. The role of N fertilization here is not clear mainly because there is no distinction between N treatments (al points are the same). The authors should show where the high N-addition-treatment points are positioned in this graph to make their explanation convincing.

Page 13, lines 16-21. This section does not provide a clear view of some potential mechanisms involved in the N saturation effect. I think the authors need either to make a more convincing case for a reduction of soil microbial respiration under N additions.

Conclusions need to be rewritten after a better interpretation of key results.

---

## Referee Comment (RC2) · Anonymous Referee #2 · 8 Dec 2016

Song and co-authors investigate how changes in N deposition affect the net CO2 sink or source strength of an alpine meadow, and study the mechanisms that govern changes in CO2 processes. They measure NEE and ER, soil and microbial respiration and estimate aboveground plant and root respiration in plots across a N addition gradient. I would like to highlight that only a few field experiments have studied this topic using multiple N addition rates, and that these studies are important to understand whether the sink strength of grasslands will saturate at future N deposition rates. Because N deposition is predicted to change during this century and we don't fully understand how it will impact CO2 processes in terrestrial ecosystems, the topic is of global importance and within the scope of Biogeosciences.

My main concerns are related with how the ms is written, how some of the data is interpreted, and with the fact that some conclusions are not justified by the results.

I believe that the ms would benefit if the authors could tighten some paragraphs in the Introduction. In my opinion the second and third paragraph of the introduction lack of direction and intent, and they are somehow repetitive. I think the authors should start this paragraph explaining that the response of NEE to N deposition is likely nonlinear, and that depending on how N affects the main components determining NEE (i.e. GEP and ER), ecosystems will transition from a N limited to a N saturation stage. For instance, some articles showed that GPP and NEP do not respond linearly to changes in N as ecosystems shift to a N saturation stage (e.g. Flescher et al., 2013, DOI: 10.1002/gbc.20026; Gomez et al., 2016, DOI: 10.1111/gcb.13187). Then they could explain how changes in N affect these main components that determine NEP through changes in processes such as plant and root biomass.

The authors added six levels of N. However, ecosystems are receiving natural rates of N deposition. Thus, I think it is important to state in Material and Methods that these experimental N rates are imposed to naturally occurring N deposition. In addition, could the authors explain why they use dry N addition treatments instead of wet?.

Just for clarity, I recommend the authors not to present results from Figure 4 until they have presented all results from Figure 3 (page 9, lines 5-13).

I am not sure I agree with the statement that 'the saturation response of Rabove and the declined response of Rmic in combination contributed to N saturation response of ER and the consequent saturation response of NEE in 2015' (page 10 lines 5-7)'. I think that if ER saturates as N increases, NEE would only saturate if GEP saturates. In addition, this statement should be in Discussion rather than Results.

Page 10, line 8-9. It is not clear to me if increased pH reduction as N increases, reduces Rmic in 2014. In both 2014 and 2015, pH decreases as N increases. Are changes in pH in 2014 affecting Rmic?

Page 10, line 10. I don't think the authors should conclude that decreased Rabove as N increased was attributed to the accumulated standing litter mass and thus less

light condition under high N addition treatments' based on a photo rather than data. In addition, this statement should not be presented in Results but in discussion.

Page 10, line 15. 'Our findings showed that ecosystem C fluxes (NEE, ER, and GEP) had linear responses in the first year but shifted to saturation responses in the second year'. Please rephrase this sentence using specific language. Based on the authors results, these processes are in the limitation stage in 2014; in 2015, they are in the limitation stage at low rates and at rates at or above 20 g N m-2 year-1 they shift to the saturation or declining stage.

The paragraph at the end of page 10, beginning of page 11 is repetitive. The first few sentences (line 15-19) are providing the same information than the last sentences (line 20-23). Please tighten the writing.

The authors state that 'saturated under N addition rate of approximately 8 gN m-2 year-1' (page 11, line 1). I think the authors are fitting thresholds 'by-eye' although there are many statistical methods that can be used to calculate thresholds.

I believe that the presentation of the idea that 'The N saturation responses of ER and thus NEE are mainly caused by the decrease of aboveground plant respiration and soil microbial respiration under high N addition treatments in 2015' (page 11) is not justified by their results. Above 15 g m-2 year-1 NEE reaches a transition threshold and it starts declining. At this stage, further N additions do not seem to be affecting Rmic (Fig. 3), and R above declines just slightly at N rates at 32 g m-2 year-1. I think the authors should consider fitting thresholds using statistical methods; this way the breaking points would be accurate and the trend of each line could be calculated. Perhaps the data that could justify this statement is in Fig. 6c. However, I think that the authors should be cautious drawing this conclusion because Rmic and RE are intrinsically correlated (i.e. Rmic is a component of RE). The authors should calculate the self-correlation coefficient instead of a simple coefficient of determination. Please see Vickers et al., 2009 (http://dx.doi.org/10.1016/j.agrformet.2009.03.009) for more information on this

statistical approach. The same applies to Rabove and Rroot, and RE; Rabove and Rroot are components of RE.

I couldn't find plant growth or standing litter biomass data that supported the statement 'The decrease of aboveground plant respiration under N32 treatment is primarily due to that N addition stimulated plant growth and thus standing litter accumulation after plant senescence' (page 11). Therefore, I am not sure this statement is justified by the authors' results. The same applies to page 14, lines 2-4.

I think that caution should be used when presenting the idea that 'Taken together with our results, it suggests that N saturation of ecosystem C fluxes may happen very quickly.' I agree with the authors that a plausible explanation could be that the net $CO_2$ sink strength of this system saturated after 2 years of treatment. However, another plausible explanation that should be acknowledged is that differences in climate between 2014 and 2015 could explain variations in the response of C fluxes to N addition. For instance, if 2015 was drier than 2014, N demands for plant growth would be met faster.

I am not sure I agree with 'Our estimate on N critical load suggests that ecosystem C cycle would be largely affected under future N deposition scenarios and ecosystem may sequester more C from the atmosphere in the alpine meadow of Qinghai-Tibetan Plateau.' because the authors conducted a 2-year study in which several levels of N were added and to present this idea I believe they would need a long-term study.

Minor comments

Page 2, line 9 – I am not sure that I agree with the statement that 'ecosystem net C sequestration is usually predicted to increase under rising N deposition'. Some articles suggest that net C sequestration will increase and others show that it will decrease. See for instance Naddelhoffer et al. 1999 (doi:10.1038/18205). Please rephrase.

Page 3, line 5 – I am not sure that 'the C cycle gets saturated', I think I would rather

prefer if the authors refer to the specific process that is saturated (e.g. the C sink strength saturates). Please rephrase.

Page 7, line 2 – I think the authors mean 'simultaneous' rather than 'contemporaneous'. Please clarify.

Page 8, line 6 – I think that the authors mean 'monthly mean NEE' rather than 'annual mean NEE'. Please rephrase throughout the ms.

Page 11, line 5 – 'a N addition gradient experiment' rather than 'an N addition experiment'.
* * *

---

## Author Comment (AC1) · 26 Jan 2017

*This study addresses potential responses of different ecosystem C fluxes to gradual increases in N fertilization. The main findings of this study provide evidence that N saturation of ecosystem C fluxes can occur in a short period of time (just over 2 years since the start of the N fertilization experiment). Key findings are shown in Fig 2 where differences in NEE and ER are clear between years and along the N fertilization treatment.*

*Despite the results indicate that N saturation may occur at increasing N fertilization levels, the underlying mechanisms explaining why C fluxes might get saturated with N inputs are not clear. The authors suggest that decreases in NEE and ER under greater N fertilization are due to decreases in plant aboveground respiration and soil microbial respiration. Looking at Figs 3 and 4, this interpretation is not really supported by results whereby plant aboveground respiration (in 2015; Fig 4a) seems to increase rather than decrease at N8,16,32 treatments compared to N0,2,4. Similarly soil microbial respiration does not seem to decrease much under N8,16,32 treatments (Fig 3d) and actually might increase under N32 compared to N16. My point here is that although NEE and ER trends are relatively clear, the mechanisms invoked here to explain these changes are not really supported by the results. There is a problem with results interpretation here that the authors need to deal with (see my comments below).*

**Response: We appreciate the reviewer very much for the thoughtful comments. We address these specific comments below, and please note that our responses are bolded. We agree with the reviewer that the mechanisms should be demonstrated more clearly. Above all, we should state that the decreases of plant aboveground respiration and soil microbial respiration ($R_{mic}$) under the highest N addition rate were compared to that under N saturation point rather than the control treatment. We are sorry about the confused statements in the previous MS, and have explained it more clearly in the revised MS.**

**From the following Fig. R1k (Fig. 4e in the previous version of the MS), we can see that plant aboveground respiration decreased under N32 compared to N16. More importantly, only $R_{mic}$ showed distinctively inverse responses to N addition rates between years, which kept increasing in 2014 (Fig. R1c) but decreasing in 2015 (Fig. R1i) along the N addition gradient. $R_{mic}$ did decline under N32 in 2015, and soil acidity under similar N addition rate was also indicated to be the reason why $R_{mic}$ decreased in grasslands (Chen et al., 2016; Liu et al., 2014). All these points have been clarified for better results interpretation.**

Chen    D, Li J, Lan Z, Hu S, Bai Y (2016) Soil acidification exerts a greater control on soil respiration than soil nitrogen availability in grasslands subjected to long-term nitrogen enrichment. Functional Ecology, 30, 658–669.

**Liu W, Jiang L, Hu S, Li L, Liu L, Wan S (2014) Decoupling of soil microbes and plants with increasing anthropogenic nitrogen inputs in a temperate steppe. Soil Biology and Biochemistry, 72, 116-122.**

[Figure]

**Fig. R1 Ecosystem respiration (ER) (a, g) and its components in response to the N addition gradient in 2014 and 2015 (mean ± SE, n = 5). SR: soil respiration (b, h), $R_{mic}$: soil microbial respiration (c, i), $R_{plant}$: plant respiration (d, j), $R_{above}$: aboveground plant respiration (e, k), $R_{root}$: plant root respiration (f, l). N0, N2, N4, N8, N16, N32 represent N addition rate is 0, 2, 4, 8, 16, 32 gN $m^{-2}$ $year^{-1}$, respectively.**

*I think the authors should either better demonstrate that soil microbial respiration might play a role in mediating the N saturation effect or that other mechanisms are at play. It looks like that soil respiration in general decreases more convincingly under higher N treatments than soil microbial respiration. Also the explanation that greater standing litter might reduce plant aboveground respiration through reduced light availability makes sense but is not really supported by the results in Fig 4e for example.*

**Response: The decrease in soil respiration (SR) in 2015 was apparently caused by decrease in $R_{mic}$. The reduction of $R_{mic}$ under high N addition level, together with low root respiration, resulted in decrease of SR in 2015. In 2014, increase of $R_{mic}$ partly offset by the decrease of root respiration, and as a result, SR had no significant difference among N treatments.**

**In 2014, plant aboveground biomass (AGB) was stimulated under high N addition treatment, especially AGB of grasses (Fig. R2). In this grassland, grasses usually have higher height than other plants. The accumulation of grasses standing litter under high N addition treatment limited light condition for other plants and negatively influenced plant growth in early growing season in 2015. We have added Fig. R2 to the revised MS, which will demonstrate our results more clearly.**

[Figure]

**Fig. R2 Plant aboveground biomass in response to the N addition gradient in 2014. N0, N2, N4, N8, N16, N32 represent N addition rate is 0, 2, 4, 8, 16, 32 gN m$^{-2}$ year$^{-1}$, respectively.**

*Also in relation to results interpretation, the authors need to acknowledge that variability in their findings could be related to their very short-term study, which may not capture key changes in NEE and ER and the underlying mechanisms involved. I would expect that it will take 3-5 years of N fertilization to better clarify these.*

**Response: Thanks for the critical comments. Ecosystem C fluxes may respond to N addition in different ways during different stages and the underlying mechanisms may also change, just as the N saturation theory stated. Although it is better to take a long-term study to clarify the underlying mechanisms, we believe that our study found the early response signals of changes in ecosystem C fluxes under N addition and revealed the potential mechanisms at early stage.**

*Overall, the manuscript needs a thorough editing in relation to sentence structure and language especially abstract and introduction but in general all throughout the manuscript.*

**Response: Thanks for pointing this out. We have tightened some paragraphs in the Introduction as the Referee #2 suggested, and carefully edited the sentence structure and language throughout the MS.**

*Discussion*
*I am not sure whether the explanation that: "The N saturation responses of ER and thus NEE are mainly caused by the decrease of aboveground plant respiration and soil microbial respiration under high N addition treatments in 2015" (page 11, lines 8-9), is well supported by the results. If I look at Fig. 4e I see an increase in aboveground plant respiration (i.e. Rabove) in 2015 under the N16 treatment and a slight decrease under the N32 treatment, which is however still higher than the N8 treatment. What I can see is an overall decrease of Rabove across all treatments in 2015 when compared to 2014. Even the 'assumed' decreases in soil microbial respiration are not clear in Fig. 3d, actually it looks like that Rmic almost increases between N16 and N32.*
*Page 11, lines 9-11. I might agree with the statement that: "The decrease of*

*aboveground plant respiration under N32 treatment is primarily due to that N addition stimulated plant growth and thus standing litter accumulation after plant senescence (Fig. S1)", but again this is not clear from the results shown. Fig. S1 might provide evidence of litter accumulation but is this the only treatment (N32), which was associated with an increase of plant litter? What about N16?*

**Response: Thanks for the reviewer's critical comments. Please see our first and second responses above. Fig. R1 and Fig. R2 explained our results more clearly.**

*Again on pag. 11, lines 17-19, the authors suggest that: "The relationships between ER and soil microbial respiration (Fig. 6c) indicate that the decrease of microbial respiration contributes to the reduction of ER under high N addition rates in 2015", which is not really what is shown in Fig. 6c. This figure shows an overall positive relationship between Rmic and ER but this has not to do with increases in N addition rates. The role of N fertilization here is not clear mainly because there is no distinction between N treatments (al points are the same). The authors should show where the high N-addition-treatment points are positioned in this graph to make their explanation convincing.*

**Response: We thank the reviewer very much for the thoughtful comments. We would replace Fig. 6 by Fig. R3 in the new draft of our MS. In Fig. R3, open circles indicate the variables under high N addition rates. We further explored the relationships between these variables only under high N addition rates (N8, N16, N32) and found that the coefficients were larger (Fig. R4), which could make the explanation more convincing as the reviewer pointed out.**

[Figure]

**Fig. R3 Relationships between aboveground plant respiration ($R_{above}$), root respiration ($R_{root}$), soil microbial respiration ($R_{mic}$) and ecosystem respiration (ER) (a,b,c), ER and net ecosystem $CO_2$ exchange (NEE) (d) across all plots in 2014 and 2015.**

[Figure]

**Fig. R4 Relationships between aboveground plant respiration ($R_{above}$), root respiration ($R_{root}$), soil microbial respiration ($R_{mic}$) and ecosystem respiration (ER) (a,b,c), ER and net ecosystem $CO_2$ exchange (NEE) (d) under N8, N16 and N32 in 2014 and 2015.**

*Page 13, lines 16-21. This section does not provide a clear view of some potential mechanisms involved in the N saturation effect. I think the authors need either to make a more convincing case for a reduction of soil microbial respiration under N additions. Conclusions need to be rewritten after a better interpretation of key results.*

**Response: Thanks for the valuable suggestions. We have made a clear interpretation based on your comments and the corresponding results in the new draft.**

**Reply to RC2**
Anonymous Referee #2

*Song and co-authors investigate how changes in N deposition affect the net $CO_2$ sink or source strength of an alpine meadow, and study the mechanisms that govern changes in $CO_2$ processes. They measure NEE and ER, soil and microbial respiration and estimate aboveground plant and root respiration in plots across a N addition gradient. I would like to highlight that only a few field experiments have studied this topic using multiple N addition rates, and that these studies are important to understand whether the sink strength of grasslands will saturate at future N deposition rates. Because N deposition is predicted to change during this century and we don't fully understand how it will impact $CO_2$ processes in terrestrial ecosystems, the topic is of global importance and within the scope of Biogeosciences.*

**Response: We greatly appreciate the reviewer for the positive comments.**

*My main concerns are related with how the ms is written, how some of the data is interpreted, and with the fact that some conclusions are not justified by the results. I believe that the ms would benefit if the authors could tighten some paragraphs in the Introduction. In my opinion the second and third paragraph of the introduction lack of direction and intent, and they are somehow repetitive. I think the authors should start this paragraph explaining that the response of NEE to N deposition is likely nonlinear, and that depending on how N affects the main components determining NEE (i.e. GEP and ER), ecosystems will transition from a N limited to a N saturation stage. For instance, some articles showed that GPP and NEP do not respond linearly to changes in N as ecosystems shift to a N saturation stage (e.g. Flescher et al., 2013, DOI: 10.1002/gbc.20026; Gomez et al., 2016, DOI: 10.1111/gcb.13187). Then they could explain how changes in N affect these main components that determine NEP through changes in processes such as plant and root biomass.*

**Response: Thank the reviewer very much for the constructive comments and suggestions. We have revised the MS as suggested and made our points more clearly. Specifically, we tighten the introduction by merging the second and third paragraph in the Introduction. The paragraph starts with explaining that the response of NEE to N deposition is likely nonlinear, which depends on how N affects the main components. Then we illustrate how ecosystems may transfer from a N limited to a N saturation stage with increasing N input. We have explained how changes in N affect these main components that determine NEE through changes in ecological processes. The references of Flescher et al. 2013 and Gomez et al. 2016 have been cited in the revised MS.**

*The authors added six levels of N. However, ecosystems are receiving natural rates of N deposition. Thus, I think it is important to state in Material and Methods that these experimental N rates are imposed to naturally occurring N deposition. In addition, could the authors explain why they use dry N addition treatments instead of wet?*

**Response: The natural N deposition rate in Chinese grasslands has been added in Material and Methods. Because the study site has high precipitation, we applied the N fertilizer when it was raining, which can make the N fertilizer dissolved and avoid additional water application. It is sound to determine only the N effects.**

*Just for clarity, I recommend the authors not to present results from Figure 4 until they have presented all results from Figure 3 (page 9, lines 5-13).*
*I am not sure I agree with the statement that 'the saturation response of Rabove and the declined response of Rmic in combination contributed to N saturation response of ER and the consequent saturation response of NEE in 2015' (page 10 lines 5-7)'. I think that if ER saturates as N increases, NEE would only saturate if GEP saturates. In addition, this statement should be in Discussion rather than Results.*

**Response: Thanks for the valuable suggestions. We have modified these**

statements in the revised MS. The reviewer is right. Fig. 2f in the MS showed that GEP also reached saturation and had similar response to the N addition gradient as NEE.

*Page 10, line 8-9. It is not clear to me if increased pH reduction as N increases, reduces Rmic in 2014. In both 2014 and 2015, pH decreases as N increases. Are changes in pH in 2014 affecting Rmic?*

**Response: In 2014, changes in soil pH did not significantly affect $R_{mic}$.**

*Page 10, line 10. I don't think the authors should conclude that decreased Rabove as N increased was attributed to the accumulated standing litter mass and thus less light condition under high N addition treatments' based on a photo rather than data. In addition, this statement should not be presented in Results but in discussion.*

**Response: We have added data of plant aboveground biomass (Fig. R2) in the revised MS. As the reviewer suggested, we have presented these statements in the Discussion.**

*Page 10, line 15. 'Our findings showed that ecosystem C fluxes (NEE, ER, and GEP) had linear responses in the first year but shifted to saturation responses in the second year'. Please rephrase this sentence using specific language. Based on the authors results, these processes are in the limitation stage in 2014; in 2015, they are in the limitation stage at low rates and at rates at or above 20 g N m$^{-2}$ year$^{-1}$ they shift to the saturation or declining stage.*
*The paragraph at the end of page 10, beginning of page 11 is repetitive. The first few sentences (line 15-19) are providing the same information than the last sentences (line 20-23). Please tighten the writing.*
*The authors state that 'saturated under N addition rate of approximately 8 gN m$^{-2}$ year$^{-1}$' (page 11, line 1). I think the authors are fitting thresholds 'by-eye' although there are many statistical methods that can be used to calculate thresholds.*

**Response: Thanks for the thoughtful comments. We have rewritten these sentences. We stated the N saturation threshold was approximately 8 gN m$^{-2}$ year$^{-1}$ based on our N addition treatments. As limited N addition rates were applied, we think it should be cautioned to calculate a certain threshold. In the revised manuscript, we used statistical method to detect the threshold.**

*I believe that the presentation of the idea that 'The N saturation responses of ER and thus NEE are mainly caused by the decrease of aboveground plant respiration and soil microbial respiration under high N addition treatments in 2015' (page 11) is not justified by their results. Above 15 g m$^{-2}$ year$^{-1}$ NEE reaches a transition threshold and it starts declining. At this stage, further N additions do not seem to be affecting Rmic (Fig. 3), and Rabove declines just slightly at N rates at 32 g m$^{-2}$ year$^{-1}$. I think the*

*authors should consider fitting thresholds using statistical methods; this way the breaking points would be accurate and the trend of each line could be calculated. Perhaps the data that could justify this statement is in Fig. 6c. However, I think that the authors should be cautious drawing this conclusion because Rmic and RE are intrinsically correlated (i.e. Rmic is a component of RE). The authors should calculate the self-correlation coefficient instead of a simple coefficient of determination. Please see Vickers et al., 2009 (http://dx.doi.org/10.1016/j.agrformet.2009.03.009) for more information on this statistical approach. The same applies to Rabove and Rroot, and RE; Rabove and Rroot are components of RE.*

**Response: Thank the reviewer very much for the critical comments and valuable suggestions. Based on the reviewer's suggestion, we have used statistical method to calculate the breaking points. We have also tried to calculate the self-correlation coefficient between components as suggested.**

*I couldn't find plant growth or standing litter biomass data that supported the statement 'The decrease of aboveground plant respiration under N32 treatment is primarily due to that N addition stimulated plant growth and thus standing litter accumulation after plant senescence' (page 11). Therefore, I am not sure this statement is justified by the authors' results. The same applies to page 14, lines 2-4.*

**Response: Thanks for the comments. We have added a figure (Fig. R2) to justify the results.**

*I think that caution should be used when presenting the idea that 'Taken together with our results, it suggests that N saturation of ecosystem C fluxes may happen very quickly.' I agree with the authors that a plausible explanation could be that the net $CO_2$ sink strength of this system saturated after 2 years of treatment. However, another plausible explanation that should be acknowledged is that differences in climate between 2014 and 2015 could explain variations in the response of C fluxes to N addition. For instance, if 2015 was drier than 2014, N demands for plant growth would be met faster.*

**Response: Thanks for the suggestion. We totally agree with the reviewer and have refined the statement by more clearly justifying the results.**

*I am not sure I agree with 'Our estimate on N critical load suggests that ecosystem C cycle would be largely affected under future N deposition scenarios and ecosystem may sequester more C from the atmosphere in the alpine meadow of Qinghai-Tibetan Plateau.' because the authors conducted a 2-year study in which several levels of N were added and to present this idea I believe they would need a long-term study.*

**Response: We agree with the reviewer. Changes in C sequestration under increasing N deposition might need longer time to study. We have deleted the sentence.**

*Minor comments*

*Page 2, line 9 – I am not sure that I agree with the statement that 'ecosystem net C sequestration is usually predicted to increase under rising N deposition'. Some articles suggest that net C sequestration will increase and others show that it will decrease. See for instance Naddelhoffer et al. 1999 (doi:10.1038/18205). Please rephrase.*

**Response: Thanks for the valuable suggestion. The reviewer is correct. We have modified the sentence.**

*Page 3, line 5 – I am not sure that 'the C cycle gets saturated', I think I would rather prefer if the authors refer to the specific process that is saturated (e.g. the C sink strength saturates). Please rephrase.*

**Response: Greet suggestion! We have specified "the C cycle" into ecosystem productivity.**

*Page 7, line 2 – I think the authors mean 'simultaneous' rather than 'contemporaneous'. Please clarify.*

**Response: The reviewer is correct! We have clarified "simultaneous" as suggested.**

*Page 8, line 6 – I think that the authors mean 'monthly mean NEE' rather than 'annual mean NEE'. Please rephrase throughout the ms.*

**Response: Thanks. We have rephrased the term throughout the MS as suggested.**

*Page 11, line 5 – 'a N addition gradient experiment' rather than 'an N addition experiment'.*

**Response: Thank the reviewer for bringing it up. We have changed into "a N addition gradient experiment".**

---

## Author Response (AR2)

**Response Letter**

Michael Bahn
Co-Editor-in-Chief
Biogeosciences

Dear Dr. Bahn,

Thank you very much for offering us the chance to revise our manuscript "Initial shifts in nitrogen impact on ecosystem carbon fluxes in an alpine meadow: patterns and causes" (bg-2016-436). We have carefully considered the thoughtful and valuable comments and suggestions from you and the reviewers. The manuscript has been revised accordingly. Here are our detailed responses to the reviews. Please note that the comments from the reviewers are in *italics* followed by our responses in **bold** text.

Sincerely,

Shuli Niu, Professor
Synthesis Research Center of Chinese Ecosystem Research Network,
Key Laboratory of Ecosystem Network Observation and Modeling,
Institute of Geographic Sciences and Natural Resources Research, Chinese Academy of Sciences, Beijing 100101
China
Phone: 86-10-6488-8062
Fax: 86-10-6488-9399
http://sourcedb.cas.cn/sourcedb_igsnrr_cas/yw/zjrck/201303/t20130306_3787558.html

**Reply to Report #1**

*I have carefully read the manuscript and I think it still has the same issues I have raised in my first review of this manuscript. I still feel confused about the main story told here and the mechanisms invoked to explain main changes in NEE.*

**Response: Thanks for the critical comments. We are sorry about for the confusing statements. We have carefully revised the manuscript and clarified the main point and mechanisms more clearly. The specific comments are addressed below.**

*1) I understand that rates of NEE increase (i.e. more negative values) under mid levels of N additions 8 and16 g N m-2 yr-1 in 2015 (second year of experiment), which seems to be the case when looking for example at Fig. 2d (this may suggest the occurrence of some sort of 'saturation' response, whose causes are far from clear).*

**Response: Thank the reviewer very much for the thoughtful comments. The reviewer is correct. Fig.2d showed the saturation response of NEE in 2015. Since NEE is the balance between GPP and ER, we explained the cause of NEE saturation by the response of GPP and ER. In Page10 Line21, we said that "The N saturation response of NEE in 2015 was mainly attributed to the saturation responses of ER and GEP (Fig. 2)".**

*2) I am confused for example when the authors say that (page 14, line 18):"The saturation responses of NEE and ER were mainly caused by N-induced decreases in aboveground plant respiration and soil microbial respiration under high N addition rates". However on page 13 line 21 the authors stated the opposite that: "In this study, greater plant growth and aboveground biomass under N addition enhanced aboveground plant respiration and thus stimulated ER". In the results section (page 9 line 5), the authors also state that: "Rabove increased with increasing N addition rates in 2014 (Fig. 4b) but got the maximum value at N16 in 2015 (Fig. 4e). If I look at Figs 4 d and e, these figures exactly show that plant respiration increased under increasing N additions. So why do the authors in the Conclusions state the opposite?*

**Response: We appreciate the reviewer very much for the thoughtful comments and really feel sorry for the confusion. We said the "saturation responses of NEE and ER were mainly caused by N-induced decreases in aboveground plant respiration and soil microbial respiration under high N addition rates", because when comparing $R_{above}$ at high N addition rate (32 gN m$^{-2}$ yr$^{-1}$) to that under the N saturation point (16 gN m$^{-2}$ yr$^{-1}$), higher N rate addition did cause decrease in $R_{above}$ (Fig. 4e). When we compared $R_{above}$ and ER under N addition rates of 2 - 16 gN m$^{-2}$ yr$^{-1}$ with those under the control, we found that "greater plant growth and aboveground biomass under N addition enhanced aboveground plant respiration and thus stimulated ER". So, these descriptions are not really opposite. They were compared with different references. In order to avoid confusion, we have rephrased these confused sentences by emphasizing the references in the revised**

**MS (Page 14, Line 3).**

*Second, the negative trend of microbial respiration with N addition rates is not clear. If I look at Fig. 3d I don't understand why Rmic for example decreases around 16 g N m-2 yr-1 but then almost increases again under the 32 g N m-2 yr-1. Also if ER (ecosystem respiration) overall increases in 2015 and microbial respiration decreases in the same year, how could it be possible that the two data points are positively correlated in Fig. 6C? I think this graph should only show data for 2015 and not for 2014.*

**Response: We greatly appreciate the reviewer's thoughtful comments. If we analyzed the data year by year, we found that there was a negative relationship between ER and $R_{mic}$ in 2015 (see Fig. R1). Based on the reviewer's suggestion, we showed the data for 2015 and not for 2014 (revised Fig. 6).**

[Figure]

**Fig. R1 Relationships between aboveground plant respiration ($R_{above}$), root respiration ($R_{root}$), soil microbial respiration ($R_{mic}$) and ecosystem respiration (ER) (a,b,c), ER and net ecosystem $CO_2$ exchange(NEE) (d) in 2015.**

**Actually, $R_{mic}$ did not "increases again under the 32 gN $m^{-2}$ $yr^{-1}$". Both the mean values for $R_{mic}$ at 16 and 32 gN $m^{-2}$ $yr^{-1}$ were 2.97 $\mu$mol $m^{-2}$ $s^{-1}$. Please see the following column figure (Fig. R2). Thanks.**

[Figure]

**Fig. R2 Soil microbial respiration ($R_{mic}$) in response to the N addition gradient in 2015 (mean $\pm$ SE, n = 5).**

*Overall, these contradictions and the lack of clarity in showing and linking the results with main discussion and conclusions make it very difficult to understand the potential contribution of this study to present knowledge.*

**Response: We have double checked the descriptions and clarified the results by emphasizing the references when make comparisons. Now we confirmed that the linkage between the results and main discussion and conclusion are correct and consistent. Hope the reviewer found our revision satisfactory. We greatly appreciate the editor for considering our manuscript!**

*The manuscript would need more language editing. These below are some suggestions: I would rephrase the first sentence (page 1, line 10) as:*
  *"Increases in nitrogen (N) deposition can greatly stimulate ecosystem net carbon (C) sequestration through positive N-induced effects on plant productivity".*
*Would rephrase second sentence (page 1, line 11) as: "However, how net ecosystem CO2 exchange (NEE) and its components might respond to increases in N deposition remains unclear".*
*Last sentence in abstract could be changed to: "Our findings bring evidence of short-term responses to increases in N deposition, which should be considered when predicting long-term changes in ecosystem net C sequestration".*

**Response: Thank the reviewer very much for the valuable suggestions. These sentences in the abstract has been rephrased according to the suggestions (Page 1, Line 11-13; Page 2, Line 3-5).**

*Page 3 lines 16, change to: "It is not clear when and how ecosystem C fluxes…"*
*First sentence of Discussion should be changed to: "Our results show that initial ecosystem C fluxes (NEE, ER, and GEP) in 2014 suggest ecosystem N limitation, whereas in 2015 these C fluxes clearly suggest N saturation under high N addition rates".*
*Lines 9-12, page 10, needs to be changed to: "These findings not only confirm the N saturation hypothesis for the response of NPP to N addition (Aber et al., 1998; Aber et al., 1989; Lovett and Goodale, 2011), but also provide comprehensive evidence of potential relationships between various ecosystem C fluxes and ecosystem N dynamics".*
*Line 12, page 10: remove "most'*
*Line 14, page 10, change to: "Using one level of N addition only might not be enough to capture and quantify complex ecosystem responses to N addition…".*

**Response: Thank the reviewer very much for the valuable suggestions. We have changed these sentences as suggested (Page 3, Line 19; Page 10, Line 11-15, 18).**

**Reply to Report #2**

*This study assesses the effect of nitrogen addition on ecosystem CO2 fluxes across a nitrogen addition gradient in an alpine meadow. Hence, this set up allows for testing*

*nonlinear effects and for saturation of responses to N addition. By inserting collars into the soil, authors also partitioned soil respiration into root and microbial respiration, addition to the process understanding of responses to N addition.*

*Before this manuscript can be accepted for publication, I think some aspects need to be improved.*

*1. Deep versus shallow collars were used to partition root from microbial respiration. This is a common technique to use for this purpose, but just like other partitioning techniques, it does have some limitations. This should be discussed in the manuscript and also its implications for the observed responses of Rmic and Rroot need to be discussed. For example, changes in plant C allocation in response to N addition can have a strong effect on Rmic, but are excluded by the deep collar method for Rmic assessment. Hence, authors could argue that Rmic responses are not solely plant-mediated.*

**Response: Thank the reviewer very much for the constructive comments and suggestions. We have added some discussion about the limitations of the partitioning technique we used (Page 14, Line 15-20).**

*2. The pH effects are overstated. A regression for Rmic versus pH is the only indication for a pH effect on Rmic, but this relationship is not necessarily causal. If authors would have had an indicator for N availability (e.g. NH4 and NO3 concentrations in soil), they would have found a strong relationship with Rmic too. To find out if pH is really a potential driver of Rmic, one would have to include multiple potential drivers (like NH4 and NO3 concentrations) to test which of the drivers best predicts Rmic.*

**Response: The reviewer is right. Soil $NH_4^+$ and $NO_3^-$ may also be drivers of $R_{mic}$. We re-analyzed the data and found that the correlation coefficient of the relationship between $\triangle R_{mic}$ and soil $\triangle pH$, $\triangle NH_4^+$ and $\triangle NO_3^-$ was 0.77, 0.68, and 0.76, respectively. So, soil pH was the most important factor driving changes in $R_{mic}$. Previous studies with similar N addition gradient also suggested that soil pH was the most important driver for responses of microbes under high N addition rates (Liu et al., 2014; Chen et al., 2016).**

**Chen   D, Li J, Lan Z, Hu S, Bai Y (2016) Soil acidification exerts a greater control on soil respiration than soil nitrogen availability in grasslands subjected to long-term nitrogen enrichment. Functional Ecology, 30, 658–669.**

**Liu W, Jiang L, Hu S, Li L, Liu L, Wan S (2014) Decoupling of soil microbes and plants with increasing anthropogenic nitrogen inputs in a temperate steppe. Soil Biology and Biochemistry, 72, 116-122.**

*3. Statistical analyses can be improved. Instead of averaging values over the year to assess the treatment effect, a linear mixed model with time as a random factor would be more appropriate.*

*Also the use of R2 as a criterion for selecting a linear or a quadratic function is not the best. R2 does not penalize overfitting, and thus gives a slight advantage to the quadratic function. Using for example AIC or BIC avoids this problem.*

**Response: We used repeated-measures ANOVA to examine N addition effects on the ecosystem C fluxes over the growing season in each year. The statistic results show as below.**

**Table R1 Results (*P* values) of repeated-measures ANOVA on the effects of nitrogen addition on ecosystem C fluxes in 2014 and 2015. NEE: net ecosystem $CO_2$ exchange, ER: ecosystem respiration, GEP: gross ecosystem production, SR: soil respiration, $R_{mic}$: soil microbial respiration.**

|      | NEE   | ER    | GEP   | SR    | $R_{mic}$ |
|------|-------|-------|-------|-------|-----------|
| 2014 | 0.020 | 0.033 | 0.002 | 0.209 | 0.246     |
| 2015 | 0.059 | 0.006 | 0.038 | 0.010 | 0.259     |

**As suggested by the reviewer, we used AIC method to re-analyze data and found the same results. Specifically, quadratic function works better than linear ones for ecosystem C fluxes in 2015. While in 2014, linear function works better than quadratic ones except SR. Please see the table below.**

**Table R2 Comparisons of Akaike information criterion (AIC) among functions describing the relationships between NEE, ER, GEP, SR and $R_{mic}$ (Y) and N addition rate (X). NEE: net ecosystem $CO_2$ exchange, ER: ecosystem respiration, GEP: gross ecosystem production, SR: soil respiration, $R_{mic}$: soil microbial respiration.**

| Functions in 2015 | Linear[1] | Quadratic[2] |
|-------------------|-----------|--------------|
| NEE               | 88.69     | 84.82        |
| ER                | 90.12     | 82.30        |
| GEP               | 87.69     | 77.68        |
| SR                | 79.43     | 78.18        |
| $R_{mic}$         | 85.15     | 84.48        |

| Functions in 2014 | Linear[1] | Quadratic[2] |
|-------------------|-----------|--------------|
| NEE               | 78.39     | 80.26        |
| ER                | 71.68     | 73.48        |
| GEP               | 77.96     | 79.86        |
| SR                | 87.88     | 87.34        |
| $R_{mic}$         | 78.33     | 80.27        |

**1) linear model:** $Y = b_1 + b_2 \times X$     **2) quadratic model:** $Y = b_1 + b_2 \times X + b_3 \times X^2$

*Specific comments:*
*bottom p2 - top p3: this needs rephrasing. Authors say that NEE may respond nonlinearly to a N addition gradient because GPP and ER can respond nonlinearly, but that is not totally correct. If GPP and ER respond nonlinearly in the same way, NEE does not change at all.*

**Response: Thanks for the suggestion. We have rephrased the relevant sentence and stated it more clearly (Page 3, Line 2-4).**

*p3, l15: 'at which time' should be 'at what N level' I suppose*

**Response: Thanks. We have rephrased this sentence (Page 3, Line 19).**

*p 5, l7: was the same amount of N given each month?*

**Response: Yes. We have clarified this in the revised MS (Page 5, Line 12).**

*p6, l16: CO2 fluxes in deep collars represent a proxy for Rmic (instead of Rmic) - see also earlier comment.*
*p11, l14: authors state that the decline in Rmic was primarily due to the pH effect, but this statement is not well supported as no other potentially important factors were assessed. I suggest authors read for example Janssens et al (2010, Nature Geoscience) to find out about other potential reasons for Rmic to decline in response to N addition. One other possibility for reduction of Rmic in response to N addition could be a shift towards more C efficient but N demanding microbial species (Agren et al 2001, Oecologia).*

**Response: Thank the reviewer very much for the thoughtful suggestions. Please see the above responses in detail. We have compared the effects of potential factors of soil pH, $NH_4^+$, and $NO_3^-$ in driving $R_{mic}$.**

**Based on the reviewer's suggestion, we have added some discussion about the potential impact of soil microbial communities on $R_{mic}$ (Page 11, Line 16-17). We are sorry that we did not monitor changes of microbial community in this study. Conducting a field experiment is very tough in the Tibet Plateau with the altitude of 3600m, but we will do that in the future.**

*p 13: authors compare their results with those of other studies. In this, they totally ignore the initial N availability as well as presence of N fixers. Both are essential though to understand differences in N effects, N saturation and the reasons behind it. I assume N availability was not measured in this experiment, or in others, but authors should recognize its importance and make readers aware that N availability needs to be assessed to further improve the understanding (differences in) responses to N addition.*

**Response: Thanks for the comments. N availability was measured in our study and it increased linearly with the N addition gradient. Please see the figure below (Fig. R3).**

[Figure]

**Fig. R3 Soil inorganic nitrogen (SIN, including $NH_4^+$, and $NO_3^-$) in response to the N addition gradient in 2015.**

*Last, although the manuscript reads quite well, it still contains quite a number of grammatical errors (like missing articles in several places) and weird phrasings. Thorough language editing is thus needed.*

**Response: Thanks for the suggestion. We have asked a native English speaker to edit the language throughout the paper.**

[revised manuscript text omitted]

---

## Author Response (AR3)

**Response Letter**

Michael Bahn
Co-Editor-in-Chief
Biogeosciences

Dear Dr. Bahn,

Thank you very much for offering us the chance to revise our manuscript "Initial shifts in nitrogen impact on ecosystem carbon fluxes in an alpine meadow: patterns and causes" (bg-2016-436). We have carefully considered the valuable comments and suggestions from you and the reviewer. The manuscript has been revised accordingly. Here are our detailed responses to the reviews. Please note that the comments from the reviewers are in *italics* followed by our responses in **bold** text.

Sincerely,

Shuli Niu, Professor
Synthesis Research Center of Chinese Ecosystem Research Network,
Key Laboratory of Ecosystem Network Observation and Modeling,
Institute of Geographic Sciences and Natural Resources Research, Chinese Academy of Sciences, Beijing 100101
China
Phone: 86-10-6488-8062
Fax: 86-10-6488-9399
http://sourcedb.cas.cn/sourcedb_igsnrr_cas/yw/zjrck/201303/t20130306_3787558.html

**Reply to Report #1**

*I reviewed this paper previously, but am not satisfied with the responses of the authors. The following points that I raised previously were not incorporated in an appropriate way.*

**Response: We feel very sorry that the reviewer was not satisfied with our revision. In this version, we have tried our best to carefully study the reviewer's suggestions and revised the manuscript accordingly. The specific comments are addressed below. Hope the reviewer is satisfied with our revision this time.**

*1) The partitioning issue is mentioned (last lines of p14), but in a way that doesn't clearly indicate how it may have influenced the results. I have the feeling that the authors did not really understand my suggestion that the response of Rmic from these collars that exclude roots indicates the N response of Rmic in the absence of plant C inputs. This could be used as an advantage (mechanistic insight into microbial responses in the absence of plant effects) but instead authors only defend their approach by saying that it has been used in many other studies.*

**Response: Thank the reviewer for the thoughtful comments. We have added some discussions about the limitations of the approach we used in the Discussion (Page 14, Line 20- Page 15, Line 4).**
**"This approach cuts roots and excludes effects of changes in plant C allocation on microbial respiration. Soil moisture content may also change in the deep collars, which likely affects microbial respiration. However, this method is at an advantage to explore mechanisms of microbial responses in the absence of plant effects, which is a common and useful technique to partition the components of ER and widely used in previous studies (Wan et al., 2005; Zhou et al., 2007)"**

*On the other hand, potential artifacts should also be acknowledged. I wonder if the negative relationship between Rmic and N addition in year 2 may be related to the increase of Rmic with N addition in year 1. Is it possible that microbes changed from being N limited in year 1 to being C limited in year 2 - something that may not be the case in the presence of plant C inputs.*

**Response: The reviewer is right. The results may be influenced by the approach we used. However, we do not think $R_{mic}$ is C limited in the second year. First, both the control treatment and the N addition treatment excluded plant C inputs, the negative relationship between $R_{mic}$ and N addition could be driven by other factors besides C inputs. Second, plant biomass increased under N addition treatments in year 2, so the supply of C was not decreasing.**

*2) More importantly: the authors' statement that pH is driving Rmic responses is not well supported. Previously, authors had not tested any other variable, and just used the relationship between Rmic and pH to argue that pH was driving the Rmic response. In the revised version, they made 2 extra correlations: Rmic vs NH4 and Rmic vs NO3.*

*Correlation coefficients were 0.77 for pH effect, 0.68 for NH4 effect and 0.76 for NO3 effect. This minor difference in correlation coefficients is then used to support that pH was most important, but in my view, it is not obviously better than NO3 (and I think NO3 + NH4 should have been tested too – but in a better approach). More decent statistics (e.g. stepwise regression analysis) would be needed to separate the pH effect from the NO3 effect, and if these cannot be distinguished, it should at least be shown and discussed*

**Response: We greatly appreciate the reviewer's thoughtful comments. We used stepwise regression analysis to re-analyze the data and found the same result. Only soil pH was left in the model. Thus, the result of stepwise regression analysis confirmed that soil pH was the most important factor driving changes in $R_{mic}$.**

*3) I asked to use AIC instead of R2 to compare goodness of fit for quadratic vs linear functions. While authors indicate the AIC results in the response letter, they do not use it in the manuscript but use again R2. Even if the results are the same, this needs to be corrected.*

**Response: Thanks for the valuable suggestions. We have added the AIC results as supplementary material in the revised paper.**

*4) I asked to improve the comparison with other experiments by considering the fact that initial nutrient availability may have been very different, and responses likely differ in nutrient-poor vs nutrient-rich systems. They did not do this, and in the response letter only show that they measured NH4 and NO3 and that it increased with N addition. This is not what I asked for. When comparing the N response of different ecosystems, the original N status should be considered. An infertile system likely responds very differently than a fertile system. If the fertility of other systems (to which the current study is compared) is unknown, authors should at least mention the relevance of it.*

**Response: Thank the reviewer very much for the thoughtful suggestions. We have added some discussions about the initial nutrient availability in the Discussion (Page 13, Line 9-10).**
**"The differences could also relate to the initial nutrient availability in different sites. Ecosystems with high N availability may reach N saturation at low rate of N addition, if there are no other limited factors."**

*In addition, also language editing is still needed. Even though authors say the manuscript has been checked by a native English speaker, I found several grammatical errors (e.g. missing articles in several places, just like in the previous version )*

**Response: Thanks for the suggestion. We have carefully checked the language throughout the manuscript.**

[revised manuscript text omitted]

---

## Author Response (AR4)

**Response Letter**

Michael Bahn
Co-Editor-in-Chief
Biogeosciences

Dear Dr. Bahn,

Thank you very much for handling our manuscript "Initial shifts in nitrogen impact on ecosystem carbon fluxes in an alpine meadow: patterns and causes" (bg-2016-436). We have carefully revised the manuscript according to your suggestions. Here are our detailed responses.

*1) Refer to the stepwise regression and its outcome explicitly in the manuscript.*

**Response: We have referred to the stepwise regression in the Statistical analysis and Results (Page 7, Line 21; Page 10, Line 8-11).**

*2) AIC results should not just be listed in the supplement but also be explained in the manuscript.*

**Response: Thanks for the suggestion. We have added the AIC results in the main manuscript and referred to it in the Results (Table 2; Page 8-9).**

*3) The new sentences on p. 13 require some language editing:*
*a) The differences … nutrient availability at (instead of 'in') different sites.*
*b) … if there are no other limiting (instead of 'limited') factors.*

**Response: Thanks. We have rephrased these sentences (Page 13, Line 7-8, 12-13).**

Sincerely,

Shuli Niu, Professor
Synthesis Research Center of Chinese Ecosystem Research Network,
Key Laboratory of Ecosystem Network Observation and Modeling,
Institute of Geographic Sciences and Natural Resources Research, Chinese Academy of Sciences, Beijing 100101
China
Phone: 86-10-6488-8062
Fax: 86-10-6488-9399
http://sourcedb.cas.cn/sourcedb_igsnrr_cas/yw/zjrck/201303/t20130306_3787558.html

[revised manuscript text omitted]